# Statistical control for spatio-temporal MEG/EEG source imaging with desparsified multi-task Lasso

**Jerome-Alexis Chevalier**
Inria Saclay
Paris-Saclay, France
jerome-alexis.chevalier@inria.fr

**Alexandre, Gramfort**
Inria Saclay
Paris-Saclay, France
alexandre.gramfort@inria.fr

**Joseph Salmon**
IMAG, Université de Montpellier
Montpellier, France
joseph.salmon@umontpellier.fr

**Bertrand, Thirion**
Inria Saclay, CEA
Paris-Saclay, France
bertrand.thirion@inria.fr

## Abstract

Detecting where and when brain regions activate in a cognitive task or in a given clinical condition is the promise of non-invasive techniques like magnetoencephalography (MEG) or electroencephalography (EEG). This problem, referred to as source localization, or source imaging, poses however a high-dimensional statistical inference challenge. While sparsity promoting regularizations have been proposed to address the regression problem, it remains unclear how to ensure statistical control of false detections. Moreover, M/EEG source imaging requires to work with spatio-temporal data and autocorrelated noise. To deal with this, we adapt the desparsified Lasso estimator —an estimator tailored for high dimensional linear model that asymptotically follows a Gaussian distribution under sparsity and moderate feature correlation assumptions— to temporal data corrupted with autocorrelated noise. We call it the desparsified multi-task Lasso (d-MTLasso). We combine d-MTLasso with spatially constrained clustering to reduce data dimension and with ensembling to mitigate the arbitrary choice of clustering; the resulting estimator is called ensemble of clustered desparsified multi-task Lasso (ecd-MTLasso). With respect to the current procedures, the two advantages of ecd-MTLasso are that *i)*it offers statistical guarantees and *ii)*it allows to trade spatial specificity for sensitivity, leading to a powerful adaptive method. Extensive simulations on realistic head geometries, as well as empirical results on various MEG datasets, demonstrate the high recovery performance of ecd-MTLasso and its primary practical benefit: offer a statistically principled way to threshold MEG/EEG source maps.

## 1   Introduction

Source imaging with magnetoencephalography (MEG) and electroencephalography (EEG) delivers insights into brain activity with high temporal and good spatial resolution in a non-invasive way (Baillet et al., 2001). It however requires to solve the bioelectromagnetic inverse problem, which is a high-dimensional ill-posed regression problem. Various approaches have been proposed to regularize the estimation of the regression coefficients that map activity to brain locations. Historically, $\ell_2$ regularization was considered first (Hämäläinen and Ilmoniemi, 1994), with successive improvements known as dSPM (Dale et al., 2000) and sLORETA (Pascual-Marqui, 2002) that are referred to as "noise normalized" solutions. The reason is that the coefficients are standardized with an estimate

of the noise standard deviation, producing outputs that are comparable to T or F statistics, yet not statistically calibrated. These latter techniques have since become standard when using $\ell_2$ approaches.

More recently, alternative approaches based on sparsity assumptions have been proposed with the ambition to improve the spatial specificity of M/EEG source imaging (Matsuura and Okabe, 1995; Haufe et al., 2009; Gramfort et al., 2012; Lucka et al., 2012; Wipf and Nagarajan, 2009). The output of such methods consists of focal sources as opposed to blurred images obtained with $\ell_2$ regularization. However, obtaining statistics ("noise normalized") from sparse or non-linear estimators seems challenging, especially since M/EEG data are spatio-temporal data with complex noise structure. A natural way to deal with the temporal dimension is to consider a multi-task estimator and structured sparse priors based on $\ell_1/\ell_2$ mixed norms (Ou et al., 2009; Gramfort et al., 2012).

In the statistical literature, some attempts to obtain an estimate of both regression coefficients and their variance have been proposed for linear models in high dimension (Wasserman and Roeder, 2009; Meinshausen et al., 2009; Bühlmann, 2013). These estimates can then be translated to $p$-value maps, *i.e.,* maps of $p$-values associated with each covariate. Some methods adapted for sparse scenarios have then proposed to debias the Lasso to obtain $p$-values or confidence intervals (Zhang and Zhang, 2014; van de Geer et al., 2014; Javanmard and Montanari, 2014). We refer to such variants as desparsified Lasso. Recently, desparsified extensions of group Lasso have also been considered (Mitra and Zhang, 2016; Stucky and van de Geer, 2018). However, all these previous methods generally lack of power when $p \gg n$. Here, we propose to address a multi-task setting in the presence of correlated noise, and to deal with high-dimensional when $p \gg n$ leveraging on data structure as done by Chevalier et al. (2018). All these challenges need to be considered for M/EEG source imaging.

Our first contribution is to propose the desparsified multi-task Lasso (d-MTLasso), an extension of the desparsified Lasso (d-Lasso) (Zhang and Zhang, 2014; van de Geer et al., 2014) to multi-task setting (Obozinski et al., 2010). More precisely, we adapt the group formulation by Mitra and Zhang (2016) to the multi-task setting that enjoys *i)*a simple statistic test formula with *ii)*a natural integration of auto-correlated noise and *iii)*a simplification of the assumptions. Our second contribution is to introduce ensemble of clustered desparsified multi-task Lasso (ecd-MTLasso), which has two advantages compared to current methods: *i)*it offers statistical guarantees and *ii)*it allows to trade spatial specificity for sensitivity, leading to a powerful adaptive method. Our third contribution is an empirical validation of the theoretical claims. In particular, we run extensive simulations on realistic head geometries, as well as empirical results on various MEG datasets to demonstrate the high recovery performance of ecd-MTLasso and its primary practical benefit: offer a statistically principled way to threshold MEG/EEG source maps.

## 2 Theoretical Background

In this section, we give the noise model, we provide standard tools for solving the source localization problem and, mainly, we present three new methods with their assumptions and statistical guarantees.

### 2.1 Model and notation

For clarity, we use bold lowercase for vectors and bold uppercase for matrices. For any positive integer $p \in \mathbb{N}^*$, we write $[p]$ for the set $\{1, \ldots, p\}$. For a vector $\boldsymbol{\beta}$, $\boldsymbol{\beta}_j$ refers to its $j$-th coordinate. For a matrix $\mathbf{X} \in \mathbb{R}^{n \times p}$, $\mathbf{X}^{(-j)}$ refers to matrix $\mathbf{X}$ without the $j$-th column, $\mathbf{X}_{i,.}$ refers to the $i$-th row and $\mathbf{X}_{.,j}$ to the $j$-th column and $\mathbf{X}_{i,j}$ refers to the element in the $i$-th row and $j$-th column. The notation $\|\cdot\|$ refers to the Frobenius norm for matrices and to the standard Euclidean norm for vectors. For a covariance matrix $\mathbf{M}$, the Mahalanobis norm is denoted by $\|\cdot\|_{\mathbf{M}^{-1}}$ and for a given vector $\mathbf{a}$ we have $\|\mathbf{a}\|_{\mathbf{M}^{-1}}^2 \triangleq \mathrm{Tr}(\mathbf{a}^\top \mathbf{M}^{-1} \mathbf{a})$. For $\mathbf{B} \in \mathbb{R}^{p \times T}$, $\|\mathbf{B}\|_{2,1} = \sum_{j=1}^p \|\mathbf{B}_{j,.}\|$, and its (row) support is $\mathrm{Supp}(\mathbf{B}) = \{j \in [p] : \mathbf{B}_{j,.} \neq 0\}$. We assume that the underlying model is linear:

$$\mathbf{Y} = \mathbf{XB} + \mathbf{E} \ , \tag{1}$$

where $\mathbf{Y} \in \mathbb{R}^{n \times T}$ is the signal observed on M/EEG sensors, $\mathbf{X} \in \mathbb{R}^{n \times p}$ the design matrix representing the M/EEG forward model, $\mathbf{B} \in \mathbb{R}^{p \times T}$ the underlying signal in source space and $\mathbf{E} \in \mathbb{R}^{n \times T}$ the noise. We assume that there exist $\rho \in [0, 1)$ and $\sigma > 0$ such that all $t \in [T]$, $\mathbf{E}_{.,t} \sim \mathcal{N}(\mathbf{0}, \sigma^2 \mathbf{I}_n)$ and that for all $i \in [n]$ and all $t \in [T-1]$, $\mathrm{Cor}(\mathbf{E}_{i,t}, \mathbf{E}_{i,t+1}) = \rho$. For all $i \in [n]$, $\mathbf{E}_{i,.}$ is Gaussian with

Toeplitz covariance, *i.e.,* defining $\mathbf{M} \in \mathbb{R}^{T \times T}$ by $\mathbf{M}_{t,u} = \sigma^2 \rho^{|t-u|}$ for all $(t, u) \in [T]^2$, we have:

$$\mathbf{E}_{i,.} \sim \mathcal{N}(\mathbf{0}, \mathbf{M}) \ . \tag{2}$$

We further assume that $\mathbf{X}$ has been column-wise standardized and denote by $\hat{\mathbf{\Sigma}} \in \mathbb{R}^{p \times p}$ the empirical covariance matrix of $\mathbf{X}$, *i.e.,* $\hat{\mathbf{\Sigma}} = \mathbf{X}^\top \mathbf{X}/n$ with $\hat{\mathbf{\Sigma}}_{j,j} = 1$. All proofs are given in Appendix D.

## 2.2 Metrics for statistical inference in M/EEG

To quantify the ability of a M/EEG source imaging technique to obtain a good estimated $\hat{\mathbf{B}}$, a commonly reported quantity is the Peak Localization Error (PLE) (Hauk et al., 2011). It consists in measuring the distance (in mm) along the cortical surface between the true simulated source and the location with maximum amplitude in the estimator. By contrast, spatial dispersion (SD) measures how much the activity is spread out by the inverse method (Molins et al., 2008).

To quantify the control of statistical errors, we consider a generalization of the Family Wise Error Rate (FWER) (Hochberg and Tamhane, 1987): the $\delta$-FWER. As illustrated in Figure 5 in appendix, it is the FWER taken with respect to a ground truth dilated spatially by an amount $\delta$ —in the present study a distance in mm. A rigorous definition of $\delta$-FWER is given in Appendix A. The rationale is that detections made outside of the support, but less than $\delta$ away from the support should count as slight inaccuracies of the methods, not as false positives. In an analogous manner, $\delta$-FDR $= (1 - \delta$-precision) has been proposed recently as an extension of the False Discovery Rate (FDR) (Benjamini and Hochberg, 1995) to include a spatial tolerance (Nguyen et al., 2019; Gimenez and Zou, 2019). We thus characterize the selection capabilities of the methods through a $\delta$-precision/recall curve.

## 2.3 Classical Solutions

The sLORETA and dSPM estimators are derived from the ridge estimator (Hoerl and Kennard, 1970):

$$\hat{\mathbf{B}}^{\mathrm{Ridge}} = \mathbf{K}\mathbf{Y} \quad \text{where} \quad \mathbf{K} = \mathbf{X}^\top (\mathbf{X}\mathbf{X}^\top + \lambda\mathbf{I})^{-1} \ . \tag{3}$$

They are obtained by scaling each row $j$ in $\hat{\mathbf{B}}^{\mathrm{Ridge}}$ by an estimate of the noise level at location $j$. It reads (Lin et al., 2006) $\hat{\mathbf{B}}^{\mathrm{dSPM}}_{j,t} = \hat{\mathbf{B}}^{\mathrm{Ridge}}_{j,t}/\sigma_j^{\mathrm{dSPM}}$ and $\hat{\mathbf{B}}^{\mathrm{sLORETA}}_{j,t} = \hat{\mathbf{B}}^{\mathrm{Ridge}}_{j,t}/\sigma_j^{\mathrm{sLORETA}}$, where $\sigma_j^{\mathrm{dSPM}} = \sqrt{\sigma^2[\mathbf{K}\mathbf{K}^\top]_{j,j}}$ and $\sigma_j^{\mathrm{sLORETA}} = \sqrt{[\mathbf{K}(\sigma^2\mathbf{I} + \mathbf{X}\mathbf{X}^\top)\mathbf{K}^\top]_{j,j}}$. Interestingly, it can be proved that in the absence of noise and when only a single coefficient is non-zero, the sLORETA estimate has its maximum at the correct location (Pascual-Marqui, 2002). Assuming $\mathbf{B}_{.,t} \sim \mathcal{N}(\mathbf{0}, \mathbf{I})$, the covariance of $\mathbf{Y}$ reads $\sigma^2\mathbf{I} + \mathbf{X}\mathbf{X}^\top$. Hence, one can consider that sLORETA adds to dSPM an extra term in the sensor covariance matrix that comes from the sources. Note that these methods treat each time instant independently, hence ignoring source and noise temporal autocorrelations.

## 2.4 Desparsified multi-task Lasso (d-MTLasso)

Let us first recall the definition of the multi-task Lasso (MTLasso) estimator (Obozinski et al., 2010) in our setting. For a tuning parameter[1] $\lambda > 0$, it is defined as

$$\hat{\mathbf{B}}^{\mathrm{MTL}} \in \underset{\mathbf{B} \in \mathbb{R}^{p \times T}}{\operatorname{argmin}} \left\{ \frac{1}{2n} \|\mathbf{Y} - \mathbf{X}\mathbf{B}\|^2 + \lambda \|\mathbf{B}\|_{2,1} \right\} \ . \tag{4}$$

It is well known that similarly to the Lasso, MTLasso is biased: it tends to shrink rows with large amplitude towards zero. Below, we provide an adaptation of the Desparsified Lasso following the approach by Zhang and Zhang (2014), see also Mitra and Zhang (2016), to ensure statistical control. The approach relies on the introduction of score vectors $\mathbf{z}_1, \dots, \mathbf{z}_p$ in $\mathbb{R}^n$ defined by

$$\mathbf{z}_j = \mathbf{X}_{.,j} - \mathbf{X}^{(-j)}\hat{\boldsymbol{\beta}}^{(-j)}_{\boldsymbol{\alpha}_j} \ , \tag{5}$$

where, for $j \in [p]$, $\hat{\boldsymbol{\beta}}^{(-j)}_{\boldsymbol{\alpha}_j}$ is the Lasso solution (Tibshirani (1996); Chen and Donoho (1994)) of the regression of $\mathbf{X}_{.,j}$ against $\mathbf{X}^{(-j)}$ with regularization parameter[2] $\boldsymbol{\alpha}_j$. Note that these score vectors

are independent of $\mathbf{Y}$ and their computation is then equivalent to solving the node-wise Lasso (Meinshausen and Bühlmann, 2006). For such vectors, the noise model in (1) yields

$$\frac{\mathbf{z}_j^\top \mathbf{Y}}{\mathbf{z}_j^\top \mathbf{X}_{.,j}} = \mathbf{B}_{j,.} + \frac{\mathbf{z}_j^\top \mathbf{E}}{\mathbf{z}_j^\top \mathbf{X}_{.,j}} + \sum_{k \neq j} \frac{\mathbf{z}_j^\top \mathbf{X}_{.,k} \mathbf{B}_{k,.}}{\mathbf{z}_j^\top \mathbf{X}_{.,j}} \quad . \tag{6}$$

Discarding the noise term and plugging $\hat{\mathbf{B}}_{k,.}^{\mathrm{MTL}}$ as a preliminary estimator of $\mathbf{B}_{k,.}$ in (6), we coin the desparsified multi-task Lasso (d-MTLasso), a debiased estimator of $\hat{\mathbf{B}}^{\mathrm{MTL}}$ defined for all $j \in [p]$ by

$$\hat{\mathbf{B}}_{j,.}^{(\mathrm{d-MTLasso})} = \frac{\mathbf{z}_j^\top \mathbf{Y}}{\mathbf{z}_j^\top \mathbf{X}_{.,j}} - \sum_{k \neq j} \frac{\mathbf{z}_j^\top \mathbf{X}_{.,k} \hat{\mathbf{B}}_{k,.}^{\mathrm{MTL}}}{\mathbf{z}_j^\top \mathbf{X}_{.,j}} \quad . \tag{7}$$

To derive d-MTLasso statistical properties, we need the extended Restricted Eigenvalue (RE) property (Lounici et al., 2011, Assumption 3.1), detailed in Appendix B. More precisely, we assume that

(A1) $\mathrm{RE}(\mathbf{X}, s)$ is verified on $\mathbf{X}$ for a sparsity parameter $s \geq |\mathrm{Supp}(\mathbf{B})|$ and a constant $\kappa = \kappa(s) > 0$.

Roughly, A1 can be seen as a combination of sparsity and "moderate" feature correlation assumptions.

**Proposition 2.1.** *Considering the model in Equation* (1)*, assuming A1 and for a choice of $\lambda$ large enough[3] in Equation* (4)*, then with high probability:*

$$\sqrt{n}(\hat{\mathbf{B}}^{(\mathrm{d-MTLasso})} - \mathbf{B}) = \boldsymbol{\Lambda} + \boldsymbol{\Delta} \quad , \tag{8}$$

$$\boldsymbol{\Lambda}_{j,.} \sim \mathcal{N}_p(\mathbf{0}, \hat{\boldsymbol{\Omega}}_{j,j}\mathbf{M}), \text{ for all } j \in [p], \quad where \quad \hat{\boldsymbol{\Omega}}_{j,k} = \frac{n\mathbf{z}_j^\top \mathbf{z}_k}{|\mathbf{z}_j^\top \mathbf{X}_{.,j}||\mathbf{z}_k^\top \mathbf{X}_{.,k}|}$$

$$\|\boldsymbol{\Delta}\|_{2,1} = \mathrm{O}\left(\frac{s\lambda\sqrt{\log(p)}}{\kappa^2}\right) \tag{9}$$

Then, under the $j$-th null hypothesis $H_0^{(j)}$: "$\mathbf{B}_{j,.} = 0$" and neglecting the term $\boldsymbol{\Delta}$ (see Appendix D.2 for more details) in (8) as done by van de Geer et al. (2014), $\hat{\mathbf{B}}_{j,.}^{(\mathrm{d-MTLasso})}$ is Gaussian with zero-mean. Finally, using standard results on $\chi^2$ distributions (see Appendix D.1), we obtain

$$n\left\|\hat{\mathbf{B}}_{j,.}^{(\mathrm{d-MTLasso})}\right\|_{\mathbf{M}^{-1}}^2 \sim \hat{\boldsymbol{\Omega}}_{j,j}\chi_T^2 \quad .$$

If $\mathbf{M}$ is known, the quantity $n\|\hat{\mathbf{B}}_{j,.}^{(\mathrm{d-MTLasso})}\|_{\mathbf{M}^{-1}}^2/\hat{\boldsymbol{\Omega}}_{j,j}$ can be used as a decision statistic to obtain a $p$-value testing the importance of source $j$ by comparison with the $\chi_T^2$ distribution. In practice we need to estimate $\mathbf{M}$ by $\hat{\mathbf{M}}$. Notably, assuming that we have an estimator $\hat{\sigma}$ of $\sigma$ that verifies approximately $(n - \hat{s})\hat{\sigma}^2/\sigma^2 \sim \chi_{n-\hat{s}}^2$, where $\hat{s} = |\mathrm{Supp}(\hat{\mathbf{B}}^{\mathrm{MTL}})|$ (see Sec. 2.5), we take

$$\hat{f}_j := \frac{n\|\hat{\mathbf{B}}_{j,.}^{(\mathrm{d-MTLasso})}\|_{\hat{\mathbf{M}}^{-1}}^2}{T\,\hat{\boldsymbol{\Omega}}_{j,j}} \quad , \tag{10}$$

as statistic to compare with a Fisher distribution with parameters $T$ and $n - \hat{s}$, to compute the $p$-values. The full d-MTLasso algorithm is given in Algorithm 1. Note that, a Python implementation of the procedures presented in this paper is available on `https://github.com/ja-che/hidimstat` along with some examples.

## 2.5 Noise parameters estimation

In Sec. 2.1 noise is assumed homogeneous across sensors, allowing to obtain a robust estimator. Extending Reid et al. (2016) to multi-task regression, we consider the residuals $\hat{\mathbf{E}} = \mathbf{Y} - \mathbf{X}\hat{\mathbf{B}}^{\mathrm{MTL}}$, and the estimated support size $\hat{s}$. Defining, for $t \in [T]$, $\hat{\sigma}_t^2 = \|\hat{\mathbf{E}}_{.,t}\|^2/(n - \hat{s})$, an estimate of $\sigma^2$ is:

$$\hat{\sigma}^2 = \mathrm{median}(\{\hat{\sigma}_t^2, t \in [T]\}) \quad .$$

Taking the median instead of the mean avoids depending on prospective under-fitted time steps and turns out to be more robust empirically. Similarly, defining for all $t \in [T-1]$, $\hat{\rho}_t = \mathrm{cor}_n(\hat{\mathbf{E}}_{.,t}, \hat{\mathbf{E}}_{.,t+1})$ (where $\mathrm{cor}_n(.,.)$ is the empirical correlation), $\rho$ is estimated by taking $\hat{\rho} = \mathrm{median}(\{\hat{\rho}_t, t \in [T-1]\})$. Then, an estimator $\hat{\mathbf{M}}$ of $\mathbf{M}$ is given by $\hat{\mathbf{M}}_{t,u} = \hat{\sigma}^2\hat{\rho}^{|t-u|}$.

**Algorithm 1** d-MTLasso

---

**input** : $\mathbf{X} \in \mathbb{R}^{n \times p}, \mathbf{Y}$

$\hat{\mathbf{B}}^{\text{MTL}} \leftarrow \text{MTL}(\mathbf{X}, \mathbf{Y})$           `// cross-validated multi-task Lasso`

$\hat{\mathbf{E}} \leftarrow \mathbf{Y} - \mathbf{X}\hat{\mathbf{B}}^{\text{MTL}}$           `// Residuals`

$\hat{s} \leftarrow |\text{Supp}(\hat{\mathbf{B}}^{\text{MTL}})|$

**for** $t \in [T]$ **do**           `// Noise level estimation`
    $\left| \quad \hat{\sigma}_t^2 = \|\hat{\mathbf{E}}_{\cdot,t}\|^2 / (n - \hat{s}) \right.$

$\hat{\sigma}^2 = \text{median}(\{\hat{\sigma}_t^2, t \in [T]\})$

Get $\hat{\mathbf{M}}$ thanks to Sec. 2.5

**for** $j \in [p]$ **do**
    $\left| \quad \mathbf{z}_j \leftarrow \text{Lasso}(\mathbf{X}^{(-j)}, \mathbf{X}_{\cdot,j}) \right.$        `// cross-validated Lasso`

    $\left| \quad \hat{\mathbf{\Omega}}_{j,j} \leftarrow \dfrac{n \mathbf{z}_j^\top \mathbf{z}_j}{|\mathbf{z}_j^\top \mathbf{X}_{\cdot,j}||\mathbf{z}_j^\top \mathbf{X}_{\cdot,j}|} \right.$

    $\left| \quad \hat{\mathbf{B}}_{j,\cdot}^{(\text{d}-\text{MTLasso})} \leftarrow \dfrac{\mathbf{z}_j^\top \mathbf{Y}}{\mathbf{z}_j^\top \mathbf{X}_{\cdot,j}} - \sum_{k \neq j} \dfrac{\mathbf{z}_j^\top \mathbf{X}_{\cdot,k} \hat{\mathbf{B}}_{k,\cdot}^{\text{MTL}}}{\mathbf{z}_j^\top \mathbf{X}_{\cdot,j}} \right.$    `// Desparsified multi-task Lasso`

    $\left| \quad \hat{f}_j \leftarrow \dfrac{n \|\hat{\mathbf{B}}_{j,\cdot}^{(\text{d}-\text{MTLasso})}\|_{\hat{\mathbf{M}}^{-1}}^2}{T \hat{\mathbf{\Omega}}_{j,j}} \right.$        `// Inference statistics`

**return** $\hat{f}_1, \ldots, \hat{f}_p$

---

## 2.6   Clustering to handle spatially structured high-dimensional data

In the high-dimensional inference scenario considered, the number of sensors is more than one order of magnitude smaller than the number of sources, $n \ll p$. Therefore, estimators of conditional association between sources and observations struggle to identify the solution. The setting is even more difficult due to the presence of high correlation between sources (see Figure 6 in appendix). Further gains can however come from a compression of the design matrix (Bühlmann et al., 2013; Mandozzi and Bühlmann, 2016). For this we introduce a clustering step that reduces data dimensionality while leveraging spatial structure. We consider a spatially-constrained hierarchical clustering algorithm described by Varoquaux et al. (2012) that uses Ward criterion[4]. Other clustering schemes might be considered, as long as they yield spatially contiguous regions of the cortical surface. The combination of this clustering algorithm with the d-Lasso or d-MTLasso algorithms will be respectively referred to as clustered desparsified Lasso (cd-Lasso) and clustered desparsified multi-task Lasso (cd-MTLasso).

The number of clusters is denoted by $C$ and, for $r \in [C]$, we denote by $G_r$ the $r$-th group. Every cluster representative variable is given by the average of the covariates it contains. Then, reordering conveniently the columns of $\mathbf{X}$, the compressed design matrix $\mathbf{Z} \in \mathbb{R}^{n \times C}$ is given by:

$$\mathbf{Z} = \mathbf{X}\mathbf{A} \;, \quad \mathbf{A} = \begin{bmatrix} \frac{1}{|G_1|} & - & \frac{1}{|G_1|} & 0 & - & 0 & \ldots & 0 & - & 0 \\ 0 & - & 0 & \frac{1}{|G_2|} & - & \frac{1}{|G_2|} & \ldots & 0 & - & 0 \\ \vdots & \vdots & \vdots & \vdots & \vdots & \vdots & \ddots & \vdots & \vdots & \vdots \\ 0 & - & 0 & 0 & - & 0 & \ldots & \frac{1}{|G_r|} & - & \frac{1}{|G_r|} \end{bmatrix} \;, \quad (11)$$

where $\mathbf{A} \in \mathbb{R}^{p \times C}$. We say that the compression of $\mathbf{X}$ is of good quality if:

(A2) there exists $\mathbf{\Gamma} \in \mathbb{R}^{C \times T}$ such that $\mathbf{\Gamma}_{r,\cdot} = \sum_{j \in G_r} w_j \mathbf{B}_{j,\cdot}$ with $w_j \geq 0$ for all $j \in [p]$, and the associated compression loss $\mathbf{X}\mathbf{B} - \mathbf{Z}\mathbf{\Gamma}$ is "small enough" with respect to the model noise (see Appendix D.3 for more details).

(A3)[5] RE$(\mathbf{Z}, s')$ is verified on $\mathbf{Z}$ for sparsity parameter $s' \geq |\text{Supp}(\mathbf{\Gamma})|$ and constant $\kappa' = \kappa'(s') > 0$.

**Proposition 2.2.** *Assume Equation (1), A2, A3, a choice of regularization parameter in the MTLasso regression of $\mathbf{Z}$ against $\mathbf{Y}$ that is large enough, and that the largest cluster of the compression is of size $\delta$, then cd-MTLasso controls the $\delta$-FWER.*

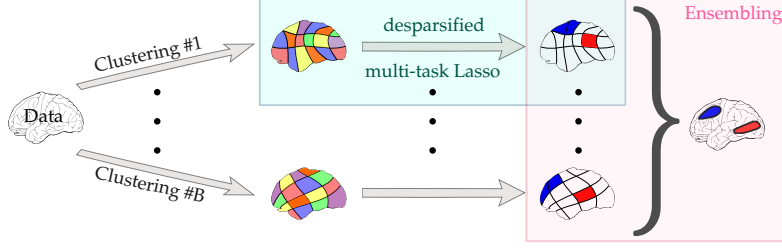

Figure 1: **ecd-MTL overview diagram.** While cd-MTLasso applies d-MTLasso to clustered data, ecd-MTLasso aggregates several cd-MTLasso solutions.

## 2.7   Ensemble of clustered desparsified multi-task Lasso (ecd-MTLasso)

To reduce the sensitivity of cd-MTLasso to small data perturbations, we propose to randomize over the clustering. We build several clustering solution, considering $B = 100$ different random subsamples of size $10\%$ of the full sample; then we aggregate the $p$-value maps output by cd-MTLasso. To aggregate the $B$ cd-MTLasso solutions, we use the adaptive quantile aggregation proposed by Meinshausen et al. (2009) detailed in Appendix C. The full procedure of ensembling $B$ cd-MTLasso (resp. cd-Lasso), solutions is called ecd-MTLasso for ensemble of clustered desparsified multi-task Lasso (resp. ecd-Lasso). Algorithm of ecd-MTLasso is given in Algorithm 2 in appendix. Also, we give an overview diagram to clarify the nesting structure of the proposed solutions in Figure 1.

**Proposition 2.3.** *Assume that for each of the $B$ compressions the hypotheses of Prop. 2.2 are verified, then ecd-MTLasso controls the $\delta$-FWER.*

This result is conservative and mixing several cd-MTLasso usually reduces the spatial tolerance $\delta$. Additional details on the procedure and computational complexity are deferred to Appendix E.

## 3   Experiments

In this section, we give empirical evidence of the advantages of ecd-MTLasso for source localization. First, in a typical point source simulation, we compare the methods with respect to the standard PLE metric; notably, we study the effect of i/clustering and ii/integrating time dimension. In a second simulation with more realistic features, we examine the $\delta$-FWER control property and compare the support recovery properties of all methods. Lastly, working on real MEG data, we show that, contrary to sLORETA, ecd-MTLasso retrieves expected patterns using a universal threshold.

### 3.1   Simulation study

Here, we study how the proposed estimators perform compared to standard $\ell_2$ regularized approaches, and assess whether time-aware statistical analysis improves upon static d-Lasso as it is essential for M/EEG source imaging. We use the head anatomy and the recording setup from the *sample* dataset publicly available from the MNE software (Gramfort et al., 2014). The design matrix $\mathbf{X}$ is computed with a three-shell boundary element model with $p = 7498$ candidate cortical locations, and a 306-channels Elekta Neuromag Vectorview system with 102 magnetometers and 204 gradiometers. We only keep the gradiometers and remove one defective sensor leading to $n = 203$. When considering multiple consecutive time instants to demonstrate the ability of the solver to leverage spatio-temporal data, the source is fixed and the temporal noise autocorrelation is set to $\rho = 0.3$.

Figure 2 reports the normalized histograms of PLE for the 7498 locations for the different methods investigated; results on spatial dispersion (SD) are available in Figure 7 in appendix. While it might seem simplistic to consider a single source, this experiment allows to demonstrate that d-Lasso improves over sLORETA in the presence of noise (see Figure 2, left). In the same figure, one can observe that clustering degrades this performance, as it carries an intrinsic spatial blur. However, even in this adversarial scenario (Dirac-like source location), cd-Lasso and ecd-Lasso remain competitive *w.r.t.* sLORETA, avoiding extreme PLE values. Note that, here, a single time point was used (T=1).

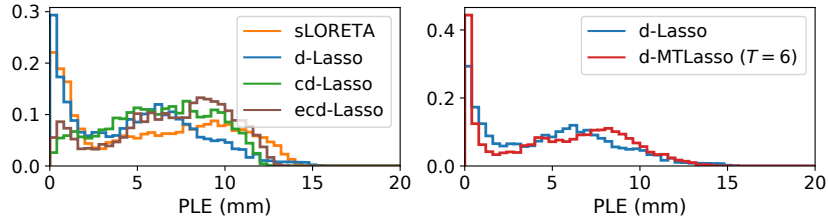

Figure 2: **Peak Localization Error (PLE) histograms.** (left): PLE on a fixed time point (T=1), sLORETA is outperformed by desparsified Lasso; cd-Lasso and ecd-Lasso are more concentrated and exhibit a smaller number of very low PLE but also a smaller number of extreme PLE values. (right): PLE for desparsified multi-task Lasso (d-MTLasso) with T=6 compared to d-Lasso (T=1). More time points improve the results by reducing the PLE.

The right panel in Figure 2 shows that d-MTLasso (T=6) significantly outperforms d-Lasso (T=1) in terms of PLE. Leveraging spatio-temporal data indeed increases the signal-to-noise ratio, which enhances spatial specificity. Effects in terms of SD are minor (see appendix, Figure 7).

## 3.2 Experiments on FWER control

We now investigate whether the different versions of d-MTLasso control the $\delta$-FWER on a realistic simulation, and compare their support recovery properties. The data are the same as in Sec. 3.1. To simulate the sources, we randomly draw 3 active regions by selecting parcels from a subdivided cortical Freesurfer parcellation with 448 parcels (Khan et al., 2018). For each selected parcel we take as sources all the dipoles at a 10-mm geodesic distance from the center of the parcel (around 10 dipoles per region), fixing the amplitude at 10 nAm. To evaluate how the methods control the $\delta$-FWER, we perform 100 simulations and count how often active sources are found outside the $\delta$-dilated ground truth. In the left panel of Figure 3, we see that d-MTLasso does not control the $\delta$-FWER, due to the violation of some hypotheses of proposition 1, in particular those regarding source correlation. However, we notice that handling noise autocorrelation reduces the empirical $\delta$-FWER. Using clustering, assumptions of Prop. 2.2 are more easily met, in particular the conditioning of the problem is improved (Mattout et al., 2005). Yet cd-MTLasso does not control the $\delta$-FWER for $\delta = 40$ mm, because the $\delta$-FWER is controlled if $\delta$ is smaller than the largest cluster diameter, which may not hold. Finally, randomization via ecd-MTLasso further improves FWER control. Empirically, we observe that the $\delta$-FWER is controlled for $\delta$ around twice the average cluster diameter. Then, with the limitation of having a compressed design matrix well conditioned ($C$ not too large), we can reduce the tolerance $\delta$ by increasing $C$ (empirical support of this claim in appendix in Figure 9). We have excluded sLORETA from this study since it does not provide guarantees on the false discoveries.

The right panel of Figure 3 shows the $\delta$-precision recall curve of the different methods. We first notice that d-MTLasso cannot compete with sLORETA, because the high dimensionality of the problem makes the computation of the source importance overly ill-posed. cd-MTLasso improves detection accuracy, but still does not perform as well as sLORETA. However, adding the ensembling step, the $\delta$-precision improves strongly, making ecd-MTLasso much better than sLORETA. In Figure 8 in appendix, we obtain similar results when considering the standard precision-recall curve.

## 3.3 Results on three MEG datasets

We now report results on three MEG datasets spanning three types of sensory stimuli: auditory, visual and somatosensory. Additional results on EEG datasets are presented in Appendix H. The auditory evoked fields (AEF) and visual evoked field (VEF) are obtained using stimuli in the left ear and left visual hemifield. The somatosensory evoked fields (SEF) are obtained following electrical stimulation of the left median nerve on the wrist. The detailed description of the data is provided in Appendix F.

Experimental results are presented in Figure 4 and Figure 10 (cf. Appendix H). Among the many methods for M/EEG source imaging present in the literature, the methods that are compared here have in common to output a statistical map. The $\ell_2$ regularized sLORETA method is compared to the

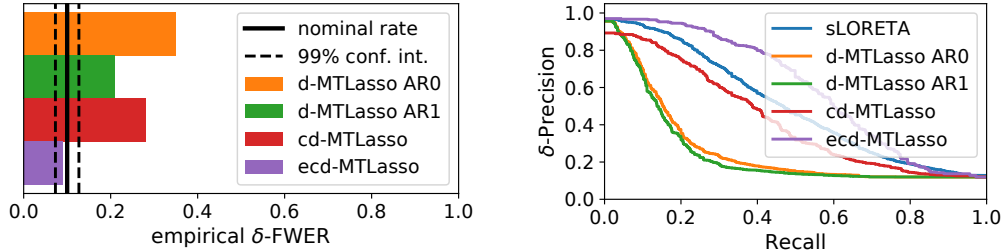

Figure 3: δ-**FWER and** δ-**Precision-Recall.** (left): δ-FWER control of the different d-MTLasso methods. δ-FWER control is hard for d-MTLasso and cd-MTLasso, as some detections are made far from the true sources, due to remote correlations. Ensembles of clusters allow to limit these false detections. (right): δ-Precision-Recall curves: sLORETA outperforms d-MTLasso AR0 and AR1, because the problem is too high dimensional for the d-MTLasso to work properly. Clustering improves the outcome, and ensembling brings further benefits: ecd-MTLasso outperforms sLORETA.

debiased sparse estimators presented and evaluated above. The input for all solvers is a time window of data: from $t = 50$ to $t = 100$ ms for AEF and VEF, and from $t = 30$ to $t = 40$ ms for SEF. During such time intervals one can expect the sources to originate primarily from the early sensory cortices whose locations are anatomically known for normal subjects.

First one can observe that all methods manage to highlight the proper functional sensory units (planum temporale for AEF, calcarine region for VEF and central sulcus for SEF). Considering sLORETA results, one can observe that at a common threshold of 3.0 on the Student statistic, the estimator is quite spatially specific for VEF, but is overly conservative for AEF and clearly leading to many false positives for SEF. By inspection of the d-MTLasso solution, one can observe that taking into account the autocorrelation of the noise leads to a better calibrated noise variance, and therefore fewer dubious detection. Considering ecd-MTLasso results, while all maps are also thresholded with a single level, one can see that it retrieves expected patterns without making dubious discoveries.

### 3.4  Summary, guidelines and limitations

**Summary of experiments.**  In Sec. 3.1, we have shown that taking into account the time dimension improve the results in terms of PLE. Also, we have seen that even in this adversarial point source scenario (cf. Sec. 3.1), clustered methods remain competitive. In Sec. 3.2, while no control of false discoveries is proposed by sLORETA, ecd-MTL is the only method that offers statistical control in practice. Namely, it controls the δ-FWER for δ equals to twice the average cluster diameter. Additionally, in this realistic simulation, ecd-MTL exhibits the best support recovery properties. In Sec. 3.3, working on real MEG data, we show that, contrary to sLORETA, ecd-MTLasso produces calibrated statistics with universal threshold and retrieves expected patterns without making dubious discoveries. Overall, ecd-MTL offers statistical guarantees and is our privileged method.

**Guidelines for statistical inference with ecd-MTLasso on temporal M/EEG data.**  First, we try to give guidelines concerning the number of clusters $C$. Hoyos-Idrobo et al. (2015) exhibit that clustering improves problem conditioning, this means that the Restricted Eigenvalue (RE) property (see assumptions A1 and A3) is more likely to be verified. Complementary, we argue that, keeping $C$ over a hundred (limiting compression loss), the fewer clusters, the more A3 is likely to be verified for Prop. 2.2 and Prop. 2.3 to hold but also the better the sensitivity of ecd-MTL. However, small $C$ also requires a higher spatial tolerance. We then hit a fundamental trade-off for statistical inference between sensitivity and spatial specificity. Then, $C$ can be chosen depending on the problem setting: if it is difficult (noisy), it seems natural to lower spatial tolerance expectations (diminish $C$); in that sense ecd-MTL is an adaptive method (cf. Figure 9). For the present use case, taking $C = 1000$ seems an adequate trade-off to ensure δ-FWER control with reasonable spatial tolerance.

Now, we give recommendation for time sampling and window size. Choosing too short windows complicate AR model estimation due to the lack of data, while choosing too large windows may lead to non stationary support. We recommend taking windows of 20 to 50ms with a time sampling at 5 to 10ms as keeping $T < 10$ reduces computation time and should not decrease sensitivity significantly.

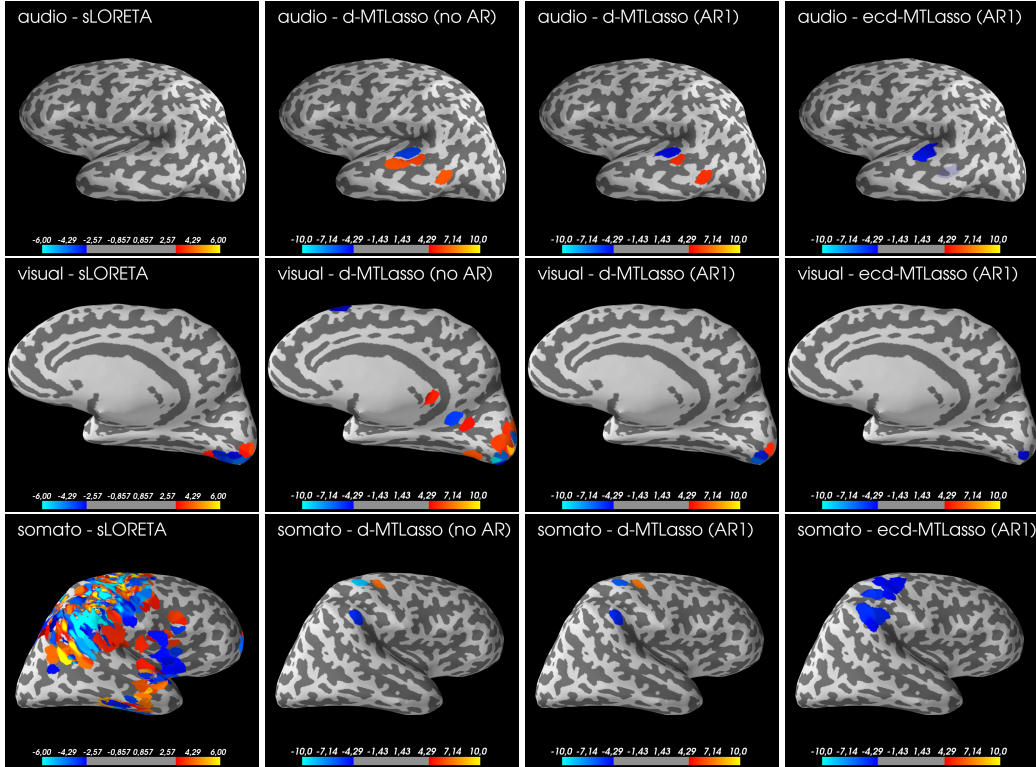

Figure 4: **Empirical comparison on 3 MEG datasets.** From left to right one can see sLORETA, d-MTLasso without AR modeling (assuming non-autocorrelated noise), d-MTLasso with an AR1 noise model and the ecd-MTLasso using also an AR1. Results correspond to auditory (top), visual (middle) and somatosensory (bottom) evoked fields. Colormaps are fixed across datasets and adjusted based on meaningful statistical thresholds in order to qualitatively illustrate FWER control issues.

Finally, when working with M/EEG data, we recommend to use only $10\%$ of the full data to compute several clustering solutions with spatial constraint and Ward criteria to ensure enough diversity.

**Limitations.** The main limitation is the fact that mixing different types of sensors violates modeling assumptions both on temporal correlations and on spatial correlations, that is why we had to treat MEG and EEG sensors separately. A possibility to handle heterogeneous sensors is to follow Massias et al. (2018b), but for the temporal part further developments are required and left for future work.

Also left for future work, is the possibility of studying windows larger than $50$ms. A simple solution is to slide a window of $20$ to $50$ms over the considered period of time.

Finally, a more common limitation is the fact that assumptions are hard to test in practice.

## 4  Conclusion

The MEG source imaging problem poses a hard statistical inference challenge: namely that of high-dimensional statistical analysis, furthermore with high correlations in the design. We have proposed an estimator that calibrates correctly the effects size and variance, up to a number of hypotheses, that are not easily met: some level of sparsity, mild correlation across sensors, homogeneity and heteroscedasticity of the noise. Up to these hypotheses, and up to a spatial tolerance on the exact location of the sources, we provide the first method with statistical guarantees for source imaging. This is made possible by bringing several improvements to the original desparsified Lasso solution: a multi-task formulation that increases power by basing inference on multiple time steps, a clustering step that renders the design less ill-posed and an ensembling step that mitigates the (hard) choice of clusters. Finally, our privileged method, ecd-MTLasso, runs in less than $10$ mn on a real dataset on non-specialized hardware, making it usable by practitioners.

# 5 Statement of broader impact

Magnetoencephalography (MEG) and electroencephalography (EEG) offer a unique opportunity to image brain activity non-invasively with a temporal resolution in the order of milliseconds. This is relevant for cognitive neuroscience to describe the sequence of active areas during certain cognitive tasks, but also for clinical neuroscience, where electrophysiology is used for diagnosis (*e.g.,* sleep medicine, epilepsy presurgical mapping). Yet, doing brain imaging with M/EEG requires to solve a challenging high-dimensional inverse problem for which statistical guarantees are crucially important. In this work, we address this statistical challenge when using sparsity promoting regularization and when considering the specificity of M/EEG signals: data are spatio-temporal and the noise is temporally autocorrelated. The proposed algorithm is built on very recent work in optimization to speed up Lasso-type solvers, as well as work in mathematical statistics on desparsified Lasso estimators. We believe that this work, whose contribution is both on the modeling side and on the inference aspects, brings sparse estimators close to a wide adoptions in the neuroscience community.

We also would like to emphasize that the inference framework can be adapted to many other high-dimensional problems where data structure can be leveraged: biomedical data and physical observations (cardiac or brain monitoring, genomics, seismology, etc.), especially those that involve severely ill-posed inverse problems.

**Acknowledgements.** This research is supported under funding of French ANR project FastBig (ANR-17- CE23-0011), the KARAIB AI chair (ANR-20-CHIA-0025-01), the European Research Council Starting Grant SLAB ERC-StG-676943 and Labex DigiCosme (ANR-11-LABEX-0045-DIGICOSME).

**Funding disclosures.** The authors certify that they have no further funding support to disclose.

## Footnotes

[1]$\lambda$ is set by cross-validation on a logarithmic grid going from $\frac{\lambda_{\max}}{100}$ to $\lambda_{\max}$, where $\lambda_{\max} = \|\mathbf{X}^\top \mathbf{Y}\|_{2,\infty}$.

[2]In (Zhang and Zhang, 2014, Table 1) an algorithm for choosing $\boldsymbol{\alpha}_j$ is proposed. We noticed that taking for all $j \in [p]$, $\boldsymbol{\alpha}_j = c\boldsymbol{\alpha}_{\max,j} := c\|\mathbf{X}^{(-j)}\mathbf{X}_{.,j}\|_\infty/n$ with $c = 0.5\%$ for M/EEG data allows to make a significant computation gain and yields adequate residuals for $C = 1000$ (see Sec. 2.6).

[3]See the proof of (Lounici et al., 2011, Theorem3.1).

[4]A typical choice is $C = 1000$ clusters for M/EEG data.

[5]$|\mathrm{Supp}(\mathbf{\Gamma})| \leq |\mathrm{Supp}(\mathbf{B})|$ and $\mathbf{Z}$ is generally better conditioned than $\mathbf{X}$ making A3 more plausible than A1.

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
