[Supplementary Material · appendix.pdf]

# APPENDIX

## A Formal definition of $\delta$-FWER control

Now we give a more formal definition of the $\delta$-FWER.

**Definition A.1** ($\delta$-family wise error rate). *Given a family of (corrected) p-values $\hat{p} = (\hat{p}_j)_{j \in [p]}$ and a threshold $x \in (0, 1)$, the $\delta$-FWER, also denoted $\mathrm{FWER}_x^\delta(\hat{p})$, is the probability to make at least one false discovery at a distance at least $\delta$ from the true support:*

$$\mathrm{FWER}_x^\delta(\hat{p}) = \mathbb{P}(\min_{j \in N^\delta} \hat{p}_j \leq x) \ , \tag{12}$$

*with $N^\delta = \{j \in [p] : \forall k \in \mathrm{Supp}(\mathbf{B}), d(j, k) \geq \delta\}$ and $d(j, k)$ is the distance between source $j$ and $k$.*

**Definition A.2** ($\delta$-FWER control). *We say that the family of (corrected) p-values $\hat{p} = (\hat{p}_j)_{j \in [p]}$ controls the $\delta$-FWER if, for all $x \in (0, 1)$:*

$$\mathrm{FWER}_x^\delta(\hat{p}) \leq x \ . \tag{13}$$

## B Extended Restricted Eigenvalue assumption

Here, we rewrite (Lounici et al., 2011, Assumption 3.1), adjusting it for the multi-task Lasso case (particular case of the more general group Lasso). Notice that for a given value of $T$, the assumption is equivalent to (Lounici et al., 2011, Assumption 4.1). Let $1 \leq s \leq p$ be an integer that gives an upper bound on the sparsity $|\mathrm{Supp}(\mathbf{B})|$. The extended Restricted Eigenvalue assumption $\mathrm{RE}(\mathbf{X}, s)$ is verified on $\mathbf{X}$ for sparsity parameter $s$ and constant $\kappa = \kappa(s) > 0$, if:

$$\min \left\{ \frac{\|\mathbf{X}\Theta\|}{\sqrt{nT} \|\Theta_J\|} : |J| \leq s, \Theta \in \mathbb{R}^{p \times T} \setminus \{\mathbf{0}\}, \|\Theta_{J^C}\|_{2,1} \leq 3 \|\Theta\|_{2,1} \right\} \geq \kappa \ , \tag{14}$$

where $J \subset [p]$ and $J^C$ denotes its complementary *i.e.,* $J^C = [p] \setminus J$, and $\Theta_J$ refers to the matrix $\Theta$ without the rows $J^C$.

## C Adaptive quantile aggregation of $p$-values and ecd-MTLasso algorithm

In this section, we provide some more details on the way we perform aggregation of $p$-values across the $p$-values maps created through the clustering randomization, then we give the full ecd-MTLasso algorithm.

For the $j$-th features (or source) we have a vector $(p_j^{(b)})_{b \in [B]}$ of $p$-values, with one $p$-value computed for each of the $B$ clusterings. Then, the final $p$-value of the $j$-th feature is given by the adaptive quantile aggregation, as proposed by Meinshausen et al. (2009):

$$p_j = \min \left\{ (1 - \log(\gamma_{\min})) \inf_{\gamma \in (\gamma_{\min}, 1)} \left( \gamma\text{-quantile} \left\{ \frac{p_j^{(b)}}{\gamma}; b \in [B] \right\} \right), \ 1 \right\} \ ,$$

where we have taken $\gamma_{\min} = 0.25$ in our experiments. Taking a value of $\gamma_{\min}$ not too small (*e.g.,* $\gamma_{\min} \geq 0.25$) allows to recover sources that have received small $p$-values several times (*e.g.,* at least for $B/4$ different choices of clustering).

We give the full algorithm of ecd-MTLasso in Algorithm 2.

## D Proofs

### D.1 Probability lemma

**Lemma D.1.** *Let $\varepsilon \in \mathbb{R}^T$ be a centered Gaussian random vector with (symmetric positive definite) covariance $\mathbf{M} \in \mathbb{R}^{T \times T}$. Then, the random variable $\varepsilon^\top \mathbf{M}^{-1} \varepsilon$ follows a $\chi_T^2$ distribution.*

**Algorithm 2** ecd-MTLasso

---

**input** $\;:\mathbf{X} \in \mathbb{R}^{n \times p}, \mathbf{Y}$

**param** $:C = 1000, B = 100$

**for** $b = 1, \ldots, B$ **do**

$\quad \mathbf{X}^{(b)} = \texttt{sample}(\mathbf{X})$
$\quad \mathbf{A}^{(b)} = \texttt{Ward}(C, \mathbf{X}^{(b)})$
$\quad \mathbf{Z}^{(b)} = \mathbf{X}\mathbf{A}^{(b)}$
$\quad q^{(b)} = \min\left\{ 1, C\, \texttt{d-MTLasso}(\mathbf{Z}^{(b)}, \mathbf{Y}) \right\}$   `// corr. cluster-wise p-val in bootstrap b`

$\quad$ **for** $j = 1, \ldots, p$ **do**
$\quad\quad \mid \;\; p_j^{(b)} = q_r^{(b)}$ if $j \in G_r$              `// corrected feature-wise p-values in bootstrap b`

**for** $j = 1, \ldots, p$ **do**
$\quad \mid \;\; p_j = \texttt{aggregation}(p_j^{(b)}, b \in [B])$        `// aggregated corrected feature-wise p-values`

**return** $p_j$ for $j \in [p]$

---

*Proof.* Note first that since $\mathbf{M}$ is symmetric positive definite, its square-root $\mathbf{N} \in \mathbb{R}^{T \times T}$ exists and is a symmetric positive definite matrix satisfying $\mathbf{N}^2 = \mathbf{M}$. Hence, this leads to the following displays

$$\boldsymbol{\varepsilon}^\top \mathbf{M}^{-1} \boldsymbol{\varepsilon} = (\mathbf{N}^{-1}\boldsymbol{\varepsilon})^\top (\mathbf{N}^{-1}\boldsymbol{\varepsilon}).$$

We have that $\mathbf{N}^{-1}\boldsymbol{\varepsilon}$ is a centered Gaussian random vector, and its covariance matrix reads:

$$
\begin{aligned}
\mathbb{E}\left[ (\mathbf{N}^{-1}\boldsymbol{\varepsilon})(\mathbf{N}^{-1}\boldsymbol{\varepsilon})^\top \right] &= \mathbb{E}\left[ \mathbf{N}^{-1}\boldsymbol{\varepsilon}\boldsymbol{\varepsilon}^\top \mathbf{N}^{-1} \right] \\
&= \mathbb{E}\left[ \mathbf{N}^{-1}\boldsymbol{\varepsilon}\boldsymbol{\varepsilon}^\top \mathbf{N}^{-1} \right] \\
&= \mathbf{N}^{-1}\mathbb{E}\left[ \boldsymbol{\varepsilon}\boldsymbol{\varepsilon}^\top \right] \mathbf{N}^{-1} \\
&= \mathbf{N}^{-1}\mathbf{M}\mathbf{N}^{-1} \\
&= \mathbf{N}^{-1}\mathbf{N}^2\mathbf{N}^{-1} \\
&= \mathrm{Id}_T \;\;.
\end{aligned}
$$

To conclude $\mathbf{N}^{-1}\boldsymbol{\varepsilon} \in \mathbb{R}^T$ is a centered Gaussian vector with covariance $\mathrm{Id}_T$, hence its squared Euclidean norm $\left\| \mathbf{N}^{-1}\boldsymbol{\varepsilon} \right\|^2 = (\mathbf{N}^{-1}\boldsymbol{\varepsilon})^\top (\mathbf{N}^{-1}\boldsymbol{\varepsilon})$ follows a $\chi_T^2$ distribution. $\qquad\square$

### D.2 Proof of Prop. 2.1

Now, we give a proof of Prop. 2.1:

*Proof.* First, let us fix an index $j \in [p]$. Then, using Equation (7) we have:

$$
\begin{aligned}
\sqrt{n}(\hat{\mathbf{B}}_{j,.}^{(\text{d}-\text{MTLasso})} - \mathbf{B}_{j,.}) &= \sqrt{n}\frac{\mathbf{z}_j^\top \mathbf{E}}{\mathbf{z}_j^\top \mathbf{X}_{.,j}} - \sum_{k \neq j} \frac{\sqrt{n}\,\mathbf{z}_j^\top \mathbf{X}_{.,k}(\hat{\mathbf{B}}_{k,.}^{\text{MTL}} - \mathbf{B}_{k,.})}{\mathbf{z}_j^\top \mathbf{X}_{.,j}} \\
&= \mathbf{\Lambda}_{j,.} + \mathbf{\Delta}_{j,.} \;\;,
\end{aligned}
\tag{15}
$$

where $\mathbf{\Lambda}_{j,.} = \sqrt{n}\frac{\mathbf{z}_j^\top \mathbf{E}}{\mathbf{z}_j^\top \mathbf{X}_{.,j}}$ and $\mathbf{\Delta}_{j,.} = \sqrt{n}\sum_{k \neq j} \mathbf{P}_{j,k}(\mathbf{B}_{k,.} - \hat{\mathbf{B}}_{k,.}^{\text{MTL}})$ with

$$\mathbf{P}_{j,k} = \frac{\mathbf{z}_j^\top \mathbf{X}_{.,k}}{\mathbf{z}_j^\top \mathbf{X}_{.,j}} \;\;.$$

Now, we show that $\mathbf{\Lambda}_{j,.} \sim \mathcal{N}_p(\mathbf{0}, \hat{\mathbf{\Omega}}_{j,j}\mathbf{M})$, or equivalently we show that $\mathbf{E}^\top \mathbf{z}_j \sim \mathcal{N}(0, n\|\mathbf{z}_j\|^2 \mathbf{M})$. It is clear that $\mathbf{E}^\top \mathbf{z}_j$ is a centered Gaussian vector. Then, its covariance denoted by $\mathbf{V}^{(j)}$, can be computed as follows:

$$\mathbf{V}^{(j)} = \mathbb{E}(\mathbf{E}^\top \mathbf{z}_j \mathbf{z}_j^\top \mathbf{E}) \in \mathbb{R}^{T \times T} \;\;,$$

whose general term is given for $t, t' \in [T]$ by

$$
\begin{aligned}
\mathbf{V}_{t,t'}^{(j)} &= \mathbb{E}(\mathbf{E}_{\cdot,t}^{\top} \mathbf{z}_j \mathbf{z}_j^{\top} \mathbf{E}_{\cdot,t'}) \\
&= \mathbb{E}(\mathbf{z}_j^{\top} \mathbf{E}_{\cdot,t'} \mathbf{E}_{\cdot,t}^{\top} \mathbf{z}_j) \qquad \text{(scalar values commute)} \\
&= \mathbf{z}_j^{\top} \mathbb{E}(\mathbf{E}_{\cdot,t'} \mathbf{E}_{\cdot,t}^{\top}) \mathbf{z}_j \\
&= \mathbf{z}_j^{\top} \mathbb{E}(\sum_{i=1}^{n} \mathbf{E}_{i,t'} \mathbf{E}_{i,t}^{\top}) \mathbf{z}_j \\
&= \mathbf{z}_j^{\top} \sum_{i=1}^{n} \mathbb{E}(\mathbf{E}_{i,t'} \mathbf{E}_{i,t}^{\top}) \mathbf{z}_j \ .
\end{aligned}
$$

Then, the noise structure in Equation (2) yields $\mathbf{V}_{t,t'}^{(j)} = \mathbf{z}_j^{\top} n \mathbf{M}_{t,t'} \mathbf{z}_j = n \|\mathbf{z}_j\|^2 \mathbf{M}_{t,t'}$.

Now, we show that with high probability $\|\boldsymbol{\Delta}\|_{2,1} = \mathrm{O}\left(\frac{s\lambda\sqrt{\log(p)}}{\kappa^2}\right)$. First, notice that:

$$
\|\boldsymbol{\Delta}\|_{2,1} \leq \sqrt{n} \max_{k \neq j} |\mathbf{P}_{j,k}| \left\| \hat{\mathbf{B}}^{\mathrm{MTL}} - \mathbf{B} \right\|_{2,1} \ .
$$

For a convenient choice of the regularization parameters $\boldsymbol{\alpha}$, using Bühlmann and van de Geer (2011, Lemma 2.1) and following the same approach as Dezeure et al. (2015, Appendix A.1), we obtain, with high probability:

$$
\sqrt{n} \max_{k \neq j} |\mathbf{P}_{j,k}| = \mathrm{O}\left(\sqrt{\log(p)}\right) \ .
$$

Bounds on $\|\hat{\mathbf{B}}^{\mathrm{MTL}} - \mathbf{B}\|_{2,1}$ are also available in the literature (Lounici et al., 2011) for $\rho = 0$ and can be extended to $\rho > 0$ similarly. Notably, provided $\rho = 0$, assuming A1 for a sparsity parameter $|\mathrm{Supp}(\mathbf{B}^*)| \leq s$, a given constant $\kappa = \kappa(s) > 0$, and a choice of $\lambda$ large enough in Equation (4), (Lounici et al., 2011, Theorem 3.1) gives directly the following bound, with high probability:

$$
\left\| \hat{\mathbf{B}}^{\mathrm{MTL}} - \mathbf{B} \right\|_{2,1} = \mathrm{O}\left(\frac{s\lambda}{\kappa^2}\right) \ .
$$

$\square$

**Remark D.1.** *Following van de Geer et al. (2014), to neglect $\boldsymbol{\Delta}$ we need to have $\|\boldsymbol{\Delta}\|_{\infty} = \mathrm{o}(1)$. This condition is verified if $s = \mathrm{o}\left(\frac{\kappa^2}{\lambda\sqrt{\log(p)}}\right)$.*

### D.3 Proof of Prop. 2.2

Before starting the proof, let us give more precision on assumption A2, the complete assumption is the following:

(A2) there exists $\boldsymbol{\Gamma} \in \mathbb{R}^{C \times T}$ such that $\boldsymbol{\Gamma}_{r,\cdot} = \sum_{j \in G_r} w_j \mathbf{B}_{j,\cdot}$ with $w_j \geq 0$ for all $j \in [p]$, so that the associated compression loss $\mathbf{XB} - \mathbf{Z}\boldsymbol{\Gamma}$ is bounded as follows:

$$
\|\mathbf{XB} - \mathbf{Z}\boldsymbol{\Gamma}\|_{2,2}^2 \leq \xi \frac{T \phi_{\min}^2(\mathbf{M})}{n} = \xi \frac{T \phi_{\min}^2(\mathbf{R}) \sigma^2}{n} \ , \tag{16}
$$

where $\xi > 0$ is an arbitrary small constant, $\phi_{\min}^2(\mathbf{M}) > 0$ is the smallest eigenvalue of $\mathbf{M}$ and $\phi_{\min}^2(\mathbf{R}) > 0$ is the smallest eigenvalue of $\mathbf{R}$, the temporal correlation matrix of the noise defined by $\mathbf{R} = \mathbf{M}/\sigma^2$. The hypothesis plainly means that the noise induced by design matrix compression is small enough with respect to the model noise.

Now we give a proof of Prop. 2.2:

*Proof.* First, we derive the d-MTLasso for the compressed problem, for $r \in [C]$:

$$\hat{\mathbf{\Gamma}}_{r,.}^{(\text{d}-\text{MTLasso})} = \frac{\mathbf{a}_r^\top \mathbf{Y}}{\mathbf{a}_r^\top \mathbf{Z}_{.,r}} - \sum_{l \neq r} \frac{\mathbf{a}_r^\top \mathbf{Z}_{.,l} \hat{\mathbf{\Gamma}}_{r,.}^{\text{MTL}}}{\mathbf{a}_r^\top \mathbf{Z}_{.,r}} \quad , \tag{17}$$

where $a_r$'s are the residuals obtained by nodewise Lasso on $\mathbf{Z}$ playing the same role as the $z_j$'s in Equation (7). Then, as done in Appendix D.2, we derive:

$$\sqrt{n}(\hat{\mathbf{\Gamma}}_{r,.}^{(\text{d}-\text{MTLasso})} - \mathbf{\Gamma}_{r,.}) = \sqrt{n} \frac{\mathbf{a}_r^\top \mathbf{E}}{\mathbf{a}_r^\top \mathbf{Z}_{.,r}} - \sum_{l \neq r} \frac{\sqrt{n}\, \mathbf{a}_r^\top \mathbf{Z}_{.,l}(\hat{\mathbf{\Gamma}}_{l,.}^{\text{MTL}} - \mathbf{\Gamma}_{l,.})}{\mathbf{a}_r^\top \mathbf{Z}_{.,r}} + \frac{\sqrt{n}\, \mathbf{a}_r^\top (\mathbf{XB} - \mathbf{Z}\mathbf{\Gamma})}{\mathbf{a}_r^\top \mathbf{Z}_{.,r}}$$

$$= \mathbf{\Lambda}_{r,.}' + \mathbf{\Delta}_{r,.}' + \mathbf{\Pi}_{r,.} \quad , \tag{18}$$

We treat $\mathbf{\Lambda}'$ and $\mathbf{\Delta}'$ as in Appendix D.2, assuming that the hypotheses that are used to bound (hence, neglect) $\mathbf{\Delta}'$ are verified (notably A3).

Next, for $r \in [C]$, we want to establish that $\frac{n\|\mathbf{\Pi}_{r,.}\|_{\mathbf{M}^{-1}}^2}{T\hat{\mathbf{\Omega}}_{r,r}'}$ is negligible, *i.e.,* that $\mathbf{\Pi}$ has a negligible effect on all decision statistics, where the covariance $\hat{\mathbf{\Omega}}'$ has the following generic diagonal term:

$$\hat{\mathbf{\Omega}}_{r,r}' = \frac{n\,\|\mathbf{a}_r\|^2}{|\mathbf{a}_r^\top \mathbf{Z}_{.,r}|^2} \quad .$$

Given that

$$\|\mathbf{\Pi}_{r,.}\|_{\mathbf{M}^{-1}}^2 = \frac{n\,\|\mathbf{a}_r^\top (\mathbf{XB} - \mathbf{Z}\mathbf{\Gamma})\|_{\mathbf{M}^{-1}}^2}{|\mathbf{a}_r^\top \mathbf{Z}_{.,r}|^2} \tag{19}$$

$$\leq n \frac{\|\mathbf{a}_r^\top\|^2}{|\mathbf{a}_r^\top \mathbf{Z}_{.,r}|^2} \frac{\|\mathbf{XB} - \mathbf{Z}\mathbf{\Gamma}\|_{2,2}^2}{\phi_{\min}^2(\mathbf{M})} \quad , \tag{20}$$

where $\|\cdot\|_{2,2}$ denotes the spectral norm. Then, we obtain that

$$\frac{n\,\|\mathbf{\Pi}_{r,.}\|_{\mathbf{M}^{-1}}^2}{T\hat{\mathbf{\Omega}}_{r,r}'} \leq \frac{n}{T} \frac{\|\mathbf{XB} - \mathbf{Z}\mathbf{\Gamma}\|_{2,2}^2}{\phi_{\min}^2(\mathbf{M})} \leq \xi \quad . \tag{21}$$

Then, if A2 is verified for $\xi$ small enough, we can also neglect $\mathbf{\Pi}$ in front of $\mathbf{\Lambda}'$.

Then, by neglecting $\mathbf{\Pi}$ and $\mathbf{\Delta}'$, we have:

$$\sqrt{n}(\hat{\mathbf{\Gamma}}^{(\text{d}-\text{MTLasso})} - \mathbf{\Gamma}) \sim \mathcal{N}_C(\mathbf{0}, \hat{\mathbf{\Omega}}_{r,r}' \mathbf{M}) \quad . \tag{22}$$

Then we can construct $p$-values that test the $r$-th null hypothesis $H_0^{(r)}$ : "$\mathbf{\Gamma}_{j,.} = 0$", applying the same technique as in Sec. 2.4. By correcting these $p$-values —*e.g.,* using the Bonferroni correction (Dunn, 1961), we multiply by $C$ the initial $p$-values—, we obtain cluster-wise corrected $p$-values that control the FWER.

Since, for all $r \in [C]$, $\mathbf{\Gamma}_{r,.}$ is a linear combination of $\mathbf{B}_{j,.}$ for $j \in G_r$, then $\mathbf{\Gamma}_{r,.} \neq 0$ if at least there exist $j \in G_r$ such that $\mathbf{B}_{j,.} \neq 0$.

Then, defining the feature-wise corrected $p$-values by the corrected $p$-values of the corresponding cluster, and assuming that clusters are at most of size $\delta$, such corrected $p$-values control the $\delta$-FWER.

$\square$

**Remark D.2.** *In assumption A2, having a positive linear combination is not necessary, a simple linear combination is sufficient.*

*However, we assumed that $\mathbf{\Gamma}_{r,.}$ was a positive linear combination of $\mathbf{B}_{j,.}$ for $j \in G_r$, to get the following desired properties:*

*"If additionally for $r \in [C]$, for all $j \in G_r$ and all $k \in G_r$, we have $\text{sign}(\mathbf{B}_{j,.}) = \text{sign}(\mathbf{B}_{k,.})$, then $\text{sign}(\mathbf{\Gamma}_{r,.}) = \text{sign}(\mathbf{B}_{j,.})$ (zero being booth positive and negative)."*

*This means that if all the features' weights in a cluster have the same sign, there exists a compression verifying A2 such that the cluster weight preserves the sign.*

### D.4 Proof of Prop. 2.3

*Proof.* Assuming the hypotheses of Prop. 2.3 and applying Prop. 2.2, we can, for each of the $B$ compression of the problem in Equation (1), construct a corrected $p$-value family that control the $\delta$-FWER. Applying the quantile aggregate method in Equation (15), we derive a corrected $p$-value family taking into account for each compression choice. Applying Meinshausen et al. (2009, Theorem 3.2), this aggregated corrected $p$-value family also controls the $\delta$-FWER. □

## E   Computational aspects

Here we give some elements about the computational aspect of the algorithms we propose.

For solving Lasso or multi-task Lasso problems, we rely for additional speed-up on `celer`[6] (Massias et al., 2018a, 2019), a solver which is much more efficient than the standard coordinate descent (speed up by more than 10x on our experiments).

To compute d-MTLasso, we must solve $p$ Lasso of size $(n, (p-1))$, and 1 multi-task Lasso with cross-validation on a dataset of size $(n, p, T)$. For $n = 200$, $p = 7500$ and $T = 10$, the algorithms can be run on a standard laptop in around 10 hours (using only 1 CPU). However, the algorithm is embarrassingly parallel and requires around 15 minutes if run on a machine with 50 CPUs. To compute cd-MTLasso, we must solve $C$ Lasso of size $(n, (C-1))$. and 1 multi-task Lasso with cross-validation on a dataset of size $(n, C, T)$. For $n = 200$, $C = 1000$ and $T = 10$, it can be run on a standard local device in less than 1 minute (using only 1 CPU). Finally, to compute ecd-MTLasso, we must solve $B$ cd-MTLasso. For $B = 100$ (25 is already a good value to get most of the advantages of ensembling), $n = 200$, $C = 1000$ and $T = 10$, it can be run on a standard laptop in around 1 hour (using only 1 CPU) and around 1 minute on a machine with 50 CPUs.

Although, when using coordinate-descent-like algorithms, the complexity depends on solver parameters such as tolerance on stopping criteria, the complexity in $C$ (or $p$) appears empirically to be cubic, while it is linear in $n$ and $T$. It is also linear in $B$.

## F   Detailed data description

For AEF and VEF, data contained one artifactual channel leading to $n = 203$, while for SEF data were preprocessed for removal of environmental noise leading to an effective number of samples of $n = 64$ (Taulu, 2006). For the AEF dataset, we report results for AEFs evoked by left auditory stimulation with pure tones of 500 Hz. The analysis window for source estimation was chosen from 50 ms to 200 ms based on visual inspection of the evoked data to capture the dominant N100m component, leading to $T = 6$. For the SEF dataset, we analyzed SEFs evoked by bipolar electrical stimulation (0.2 ms in duration) of the left median nerve. To capture the main peaks of the evoked response and exclude the strong stimulus artifact, the analysis window was chosen from 18 ms to 200 ms based on visual inspection of the sensor signal.

Preprocessing was done following the standard pipeline from the MNE software (Gramfort et al., 2014). Baseline correction using pre-stimulus data (from -200 ms to 0 ms) was used. Epochs with peak-to-peak amplitudes exceeding predefined rejection parameters (3 pT for magnetometers and 400 pT/m for gradiometers, and 150 $\mu$V for EOG on AEF and VEF and 350 $\mu$V for SEF) were assumed to be affected by artifacts and discarded. This resulted in 55 (AEF), 67 (SEF) and 111 (SEF) artifact-free measurements which were average to produce the target matrix $\mathbf{Y}$. The gain matrix was computed using a set of $p = 7498$ cortical locations, and a three-layer boundary element model.

## G   Related Work

The topic of high-dimensional inference has been addressed in many recent works. Yet, to the best of our knowledge, none of this literature has been applied to the source localization problem we consider here.

- The idea of associating clustering with high-dimensional inference can also be found in recent works with application to genetic data: Bühlmann et al. (2013) has used a fixed clustering step, which is made adaptive in Mandozzi and Bühlmann (2016). Our contribution deviates from these works in two regards: unlike Bühlmann et al. (2013), we do not consider that a fixed clustering, however good it is, indeed captures the essence of the problem: this is why we resort to an ensemble of different clustering solutions. Unlike Mandozzi and Bühlmann (2016), we do not try to narrow down the inference in a hierarchical fashion, because we do not consider that source imaging can in effect be traced down to the vertex level: given the difficulty of the source imaging problem, we find it more satisfactory to outline a region of putative activity.

- Another family of inference methods based on sample splits has been introduced by Meinshausen et al. (2009): train data are used to select regions, test data to assess their statistical significance. The choice of splits can be varied and aggregated upon to mitigate the impact of arbitrary splits selection. However, data splitting has a high cost in terms of statistical power, making these approaches weakly sensitive Taylor and Tibshirani (2015).

- An alternative method yielding family-wise error rate (FWER) control is the stability selection method, that builds on bootstrapped randomized sparse regression Meinshausen and Bühlmann (2010). Yet, this approach has been found too weakly sensitive and it has not been considered in further statistical inference works, see *e.g.,* Dezeure et al. (2015).

- Post-selection inference Taylor and Tibshirani (2015) is an approach that typically relies on a sparse estimator (such as Lasso) and then assesses the significance of the selected variables. It accounts for the selection in the inference process, avoiding the undesirable bias of selecting and testing on the same data. However, we have not not found an implementation that scales in a numerically sound way to the problem size that we are considering here: thousand features, even after clustering.

- Knockoff inference (with or without clustering) is probably the most recent alternative developed for high-dimensional inference (Barber and Candès, 2019): it consists in appending noisy copy of the problem features and selecting only variables that are much more significantly associated than their noisy copy. While this approach is computationally relevant for the problem at hand, it suffers from the arbitrary knockoff variable set used; it yields a control of the false discovery rate of the detection problem, that is not directly comparable with the family-wise error rate (FWER) considered here. FWER control is possible with knockoff Janson and Su (2016), yet very weakly sensitive.

# H   Supplementary figures

Figure 5: **Illustrating spatial tolerance of size $\delta = 20\,\text{mm}$ and $\delta = 40\,\text{mm}$.** The true source in red has a 10 mm radius (distance measured on the cortical surface) and the spatial tolerance extend this region by 20 mm on the left side and 40 mm on the right side in yellow. The $\delta$-FWER is the probability of making false discoveries outside of the extended region. Then, a false discovery made in the yellow region is not counted neither as an error nor a true positive.

Figure 6: **Illustrating correlation in MNE sample MEG data.** (left): Distribution of the maximum correlation between a feature (resp. cluster) and another connected feature (resp. cluster). (Top) the maximum connected feature correlation is close to 0.98 in average. (Bottom) the maximum connected cluster correlation is lower, close to 0.9 on average. Clustering improves conditioning significantly. (right): The density of the inter feature correlation (top) looks similar to the density of the inter cluster correlation (bottom). By focusing the extreme values of correlation, we see a little decrease of extreme values for the clustered data.

Figure 7: **Spatial Dispersion (SD) histograms.** (left): SD on a fixed time point (Hauk et al., 2011). All methods lead to comparable spatial dispersion. (right): SD for desparsified multi-task Lasso (d-MTLasso) with increasing time points. See Figure 2 for PLE histograms on the same experiments.

Figure 8: **Precision-Recall.** See Figure 3 for $\delta-$Precision-Recall curves computed on the same data.

Figure 9: **ecd-MTLasso empirical $\delta$-FWER and precision recall for different choice of cluster sizes.** (left): Running the same simulation as in Sec. 3.2, we observe that the spatial tolerance $\delta$ can be reduced to 20 mm by increasing the number of clusters up to 4000. With $C = 1000$ clusters (resp. $C = 2000$, $C = 4000$), the average cluster diameter is around 18 mm (resp. 13 mm and 9 mm). It turns out that the $\delta$-FWER is controlled for around twice the diameter (if the compressed design matrix verifies assumption A1). (right): We see that this decrease in spatial tolerance comes with a price regarding support recovery: the precision-recall curve declines with when $C$ is increased. (both): Note that we need to set the hyper-parameter $c$ that is used to compute the regularization parameters $\boldsymbol{\alpha}$ (see note coming with Equation (5)). We found empirically that it should be inversely proportional to $C$: for $C = 1000$, $c = 0.5\%$; for $C = 2000$, $c = 0.25\%$; for $C = 4000$, $c = 0.15\%$.

Figure 10: **Comparison on audio dataset on both hemispheres.** From left to right are compared sLORETA, d-MTLasso without AR modeling (noise is assumed non-autocorrelated), d-MTLasso with an AR1 noise model and the ecd-MTLasso using also an AR1. The results correspond to auditory (top) evoked fields. Colormaps are fixed across datasets and adjusted based on meaningful statistical thresholds in order to outline FWER control issues.

Figure 11: **Results on real data keeping only EEG sensors.** Auditory activations (top) have historically been hard to infer with EEG sensors: sLORETA produces only false discoveries while ecd-MTL and d-MTL make no discoveries. In the visual experiment (bottom): sLORETA and ecd-MTL produce expected patterns, d-MTL produces expected patterns plus one false discovery in the frontal lobe. In our work, we have emphasized MEG experiments: they offer more sensors compared to EEG leading to improved statistical power.

## Footnotes

[6]https://github.com/mathurinm/CELER