[Reviews · NeurIPS 2020]

Review 1

Summary and Contributions: This paper develops a desparsified multitask LASSO framework (d-MTLasso) to tackle the ill-posed inverse source localization problem that arises with EEG/MEG data. D-MTLasso models the noise covariance using an autoregressive model and uses a set of latent score vectors to regularize the relationship between the source and sensor spaces. The authors extend d-MTLasso via a clustering model and ensemble learning. These extensions provide better control over the false discovery rate. The proposed methods are compared against an existing baseline using both simulated and real-world data. The main contributions of this work are (1) combining elements of multi-task learning and desparsification into a unified d-MTLasso framework, (2) developing an efficient inference procedure for estimating the underlying spatial sources, (3) providing theoretical guarantees about controlling FDR.

Strengths: Overall, I appreciate the careful model-based approach for inverse source localization. The d-MTLasso is a simple and elegant framework that eliminates bias while encouraging sparse solutions. I also appreciate the associated decision statistic to quantify performance and the statistical guarantees on accuracy. The simulated results in Fig. 2 clearly demonstrate the potential of this method.

Weaknesses: This was a challenging paper to review, not because the methods were overly complex, but because the presentation is so convoluted. Section 2.4 (d-MTLasso) is the heart of the paper. However, it is not clear how this work differs from the desparsified formulations of Zhang (2014) and Mitra (2016) and the node-wise Lasso optimization of Meinshausen (2006), all of which are cited in the paper. The next contribution is the clustering formulation in Section 2.6. Here, it is unclear how the groups {G_r} are defined. Are they selected a priori (which seems to be the case in Section 2.7, where they are sampled randomly), or can they be optimized for better performance? In the latter case, I do not see how a clustering penalty can be incorporated into the traditional Lasso while preserving convexity. Finally, it is unclear what we take away from the real-world experiments. The d-MTLasso variants perform similarly in Fig. 3. So, how does one decide which setting to use (original, AR noise covariance, ECD)?

Correctness: The model and inference procedures seem technically sound.

Clarity: There are a few grammatical errors in the Introduction. The bigger issue is that the methods section is extremely convoluted. The d-MTLasso extensions are presented one after the other with little intuition about how they differ from the cited literature and which configuration to select in any given scenario.

Relation to Prior Work: The authors provide a detailed discussion of inverse source localization methods (Introduction). They also use some of these techniques as a starting point to the proposed work (Theoretical Background).

Reproducibility: Yes

Additional Feedback: Please refer to “weaknesses”. An overview figure that compared each extension in Section 2 would go a long way towards improving the clarity of this paper. Also, I personally do not find Proposition 2.1 very insightful. I would suggest just stating the main result and using the extra space to highlight the novelty of these methods over the cited literature and to provide guidelines about which settings to use. AFTER REBUTTAL: While the authors have put a great deal of effort into their response, I do not think it addresses my two main concerns: presentation clarity and technical innovation with respect to the cited papers. I think the authors could have done a much better job presenting their framework and highlighting its benefits. For these reasons, I will keep my rating of marginally above the acceptance threshold.


Review 2

Summary and Contributions: POST-AUTHOR-FEEDBACK-COMMENT: Thank you very much for the very professional author's feedback. The reviewer is delighted with the response and wishes to be allowed to show such feedback to students as teaching material. ---- They adapt the desparsified Lasso estimator employing an estimator of the Gaussian linear model parameter, which asymptotically follows a Gaussian distribution under sparsity and fair feature correlation assumptions. Described conniving real-world-based MEG data (data available online for reproducibility) simulations on realistic head geometries support the presented methodology.

Strengths: The authors very diligently provided a research background in the source localization field and explaining the proposed approach. The math looks convincing and MEG results are very impressive.

Weaknesses: Unfortunately the provided code does not run on the latest Python 3.8.5 and scikit-learn 0.23.1. The authors shall provide necessary requirements to run the code. Somehow results with any EEG example would be also interesting as that method is a bit superior to MEG for source localization problems.

Correctness: The author claims are nicely supported with solid experimental results. Python code looks OK even the reviewer could not run it in the latest environments (requirements explanations would help). The MEG results from an available dataset for reproducibility look very convincing.

Clarity: Yes, the paper and appendix are very clear and reading the content was a real pleasure for the reviewer.

Relation to Prior Work: The authors very dilligently provide references to prior research in the main manuscript as well as in the suplementary materials.

Reproducibility: Yes

Additional Feedback: Good job! Just Python environemnt requiriments to run smoothly the scripts would help. The reviewer failed as stated above.


Review 3

Summary and Contributions: This paper presents a computation framework, called desparsified multi-task Lasso, for source imaging using MEG/EEG data. The major contribution of this work is that the proposed method provide statistical guarantees for source imaging, which is a challenging issue in this field.

Strengths: Theoretical grounding of the proposed method is sound. The experimental results are comprehensive and convincing. The major strength of this work is the statistical calibration to detect activation regions.

Weaknesses: Experiments are conducted on 3 MEG datasets for auditory, visual, and somatosensory task, but lack of EEG datasets.

Correctness: The proposed method and experiments look correct to me.

Clarity: This paper is well written with sufficient details.

Relation to Prior Work: Prior works are clearly discussed.

Reproducibility: Yes

Additional Feedback:


Review 4

Summary and Contributions: The authors propose a method to compute the inverse problem of identifying sources of electrical brain activity from non-invasive sensors. Compared to previous methods, their formulation treats multiple time points as related thanks to a multi-task approach and includes auto-correlation in the noise pattern. They display that this technique provides a calibrated statistical output, with guarantees.

Strengths: While multiple approaches have been proposed for source reconstruction, being able to provide statistical reasoning behind the thresholding of the obtained source activations has not been satisfactorily answered (to the best of my knowledge). The statistical guarantee provided by the proposed approach is an important step in this direction. Overall, I found that this work would have impact on the EEG/MEG source reconstruction community. However other assumptions might make this work specifically tailored to EEG/MEG (e.g. spatial clustering of sources) compared to providing a general inverse reconstruction problem solution.

Weaknesses: I failed to see a flowing logic in the experiments performed and how they relate to the proposed contributions. The results of the experiments have not convinced me that ecd-MTLasso is a truly better solution compared to sLORETA or its other formulations. I think that the result around statistical testing should be emphasized in the experiments and better highlighted in the results (text). I missed a discussion of the limitations of the approach.

Correctness: I did not find obvious mistakes but was not able to deeply assess the propositions (all proofs in appendix and some context missing, e.g. RE). Comments: - Some parameters seem to be fixed (for computational efficiency?). Could you discuss the effect they might have, e.g. C=1000 clusters which seems specifically important for delta-FWER. - If we consider the simulation histograms as bimodal (peak around 0 displaying the number of true positives and distribution of distance errors), aren't cd-Lasso and ecd-Lasso pushing the error distribution to the right compared to d-Lasso, i.e. further away from the ground truth? Would the choice of C affect this shifting? - Besides histograms for Fig 1, were pairwise PLE computed across methods? Given the shifts in error distribution across methods, I am wondering whether there are differences in how PLE is affected across methods (e.g. small PLE in one method becomes 0 in another or becomes large and other sources with larger PLEs are better fitted?) - MEG data results: it is unclear to me which MTLasso solution is best. For example, the threshold seems to lead to a conservative solution for ecd-MTLasso in the visual case. - Is there an effect of the number of sensors used in the obtained solution and/or in computation times? - In the MEG experiments, time windows are quite tight. I understand that this can help identify early sensory cortices, but was wondering what the samlping rate was, how many time points were considered and whether there was an effect of the number of time points used on the obtained maps (T=1 vs T=6 in Fig 1 is a rather small comparison). - It is unclear to me how the different steps are performed (e.g. clustering/ensemble) in relation to one another and which split of the data they use.

Clarity: Overall the paper is well written. I however found some definitions rushed and some context missing. The paper might benefit from focusing on some aspects and going into more details for those. Comments: - The list of contributions is more a step by step method description. Contributions in terms of hypotheses relaxations or generalization to more cases, as well as improved performance on results should be stated instead. This step by step description would be useful in the methods section. - Assumptions could be described in a clearer way, i.e. mathematical + text. For example, assuming that the noise distribution is identical across sensors and that the auto-correlation is fixed with parameter rho. The role of the RE assumption is unclear. - It would be nice to specify which assumptions sLORETA or dSPM make and which can be relaxed given the AR definition or which additional assumptions are needed. - Equation 6: E is not defined. Does it correspond to Eq (2) or its non-AR version? It is unclear whether this is the definition of the AR multi-task desparsified Lasso or whether it is detailing de-sparsified Lasso and its MTL extension. This section feels a bit "rushed", and each step should be defined and presented more clearly. - AR0 and AR1 are not defined. minor: - typo in title (mutli) - please define "mildly high dimensional" vs "high dimensional" data

Relation to Prior Work: Prior work is discussed. From my understanding, the novelty resides in the use of multi-tasking with desparsified LASSO, with statistical guarantees for thresholding. It would be good to state this clearly in the contributions. The role of clustering and ensembling is unclear (not novel, was it added for empirical reasons, based on number of sources and variability in cluster selection?).

Reproducibility: Yes

Additional Feedback: The work will be reproducible if code is released. From the paper it would be difficult. It might be interesting to add the extra steps (clustering and ensembling) in the algorithm provided. I have read the authors' response and thank them for addressing the comments. I believe that a clearer text, with stated assumptions and limitations will improve the quality of this paper. I have amended my rating accordingly.

[Author Response · NeurIPS 2020]

We thank the reviewers for their careful reading and constructive comments. Below is a list of the actions we will take
to A) clarify certain aspects of the method description (R1/R4), B) present complementary experimental results (R2/R3)
and C) update the Python code shared in the supplementary material to solve the packaging issue (R2).

**A) Clarifications**. They concern statistical assumptions and contributions, as well as take-away from experiments.

Improvements to the "flowing logic" (R4) clarification of the "take-away" (R1) in experiments (Sec. 3). We will add an
introductory paragraph before Sec. 3.1, presenting briefly (5 lines) the sequence of experiments and their rationale. It
amounts to: 1/point spread estimates which are common practice for M/EEG source localization, 2/realistic simulations
to show $\delta$-FWER control and benchmark against competing methods, 3/results on real data.

ecd-MTL experimental benefits (R1/R4). ecd-MTL is our privileged solution as it remains competitive in an adversarial
simulation (Exp. 1, line 192-197), has best recovery properties in realistic simulations (Exp. 2), is the only method that
offers statistical control (Exp. 2), and produces statistics with universal threshold (Exp. 3) contrary to sLORETA.

Statistical guarantees thanks to randomized clustering (R1/R4). Clustering improves conditioning, which allows to
verify the Restricted eigenvalue (RE) property (assumptions A1 and A3). RE property is a standard technical assumption
(Bickel et al. 2009). The less clusters (i.e. small $C$), the more A3 is likely to be verified for Prop. 2.2 and 2.3 to hold.
Also, with smaller $C$ the sensitivity of ecd-MTL is improved. However, a small $C$ also requires higher spatial tolerance.
We then hit a fundamental trade-off for statistical inference between sensitivity and spatial specificity. This is a key
contribution of our work. Taking $C = 1000$ seems for the present use case an adequate trade-off to ensure $\delta$-FWER
control with reasonable spatial tolerance. Note that the clusters are obtained a priori (cf. Algo. 3). The clusters are
not optimized using the target data, but sampled from a set of good candidate parcellations. A key idea of ecdl-MTL
is indeed to randomize the definition of the clusters to mitigate the bias due to a specific choice of clusters. Good
clusterings are necessary to minimize the compression loss (assumption A2) while having highly variable clustering
solutions is necessary to benefit from ensembling. Altogether these considerations motivate the use of the ecd-MTL
approach over the d-MTL or the sLORETA methods that do not offer the same statistical guarantees (R1/R4).

Combining heterogeneous sensors (R2/R3/R4). Mixing different types of sensors would violate our modelling
assumptions both on temporal correlations and on spatial correlations. A possibility to handle heterogeneous sensors is
to follow Massias et al. (2018): Generalized Concomitant Multi-Task Lasso for Sparse Multimodal Regression, but for
the temporal part further developments are required and left for future work.

Testing assumptions (R4). Another limitation is the fact that assumptions A1, A2 and A3 are hard to test in practice.

"3.4 Summary, guidelines, limitations" (R1/R4). We will add a sub-section to summarize the aspects discussed above.

Theoretical novelty (R1/R4). Our work is complementary to Mitra et al. (2016): taking the Multi-Task approach allows
for a i/ simple statistic test formula, ii/ the integration of auto-correlated noise and iii/ a simplification of mathematical
assumptions since they reduce to the RE assumption. Additionally, our present work can be used for solving inverse
reconstruction problem of spatially structured data (medical imaging, genomics, geosciences, etc.) (R4).

**B) New EEG experiments** (R2/R3). Below, we present results on real data keeping only EEG sensors (added to app.).
Bilateral auditory activations have historically been hard to infer with EEG sensors: In F1-F3, sLORETA produces only
false discoveries (FD) while ecd-MTL and d-MTL make no discoveries. In the visual experiment G1-G3: sLORETA
and ecd-MTL produce expected patterns, d-MTL produces expected patterns + one false discovery in the frontal lobe.
In the paper, we have emphasized MEG experiments: with more sensors w.r.t. EEG, statistical power is improved (R4).

**C) Algorithms and Code.** (R2/R4) Our attached code runs fine with sklearn 23.1, but requires running 'celer 0.5dev'
(requirements were in the read.md file). Also, we noticed that 'pyvista', which is used for 3D visualization, has evolved
and now requires 'pyvistaqt', to use 'mayavi' instead please comment l.90 of the main script. Code will be released and
tested on GitHub. Concerning algorithmic clarity (R1/R4), we remind that Alg. 3 in app. synthesizes ecd-MTL. For
further clarity about the successive algorithmic extensions, we will add an overview diagram of the methods (R1).

(F1)          (F2)          (F3)          (G1)          (G2)          (G3)

[Meta-Review · NeurIPS 2020]

Four reviewers have found that the methodological contributions of this paper will be impactful for source reconstruction. The author response was appreciated by the reviewers, who recommend acceptance, and I agree with them. However, please make sure to address the clarity concerns brought up by R1 (which include additional changes beyond the commitment for clarity in the author response).